# impuTMAE: Multi-modal Transformer with Masked Pre-training for Missing Modalities Imputation in Cancer Survival Prediction

## Abstract

The use of diverse modalities, such as omics, medical images, and clinical data can not only improve the performance of prognostic models but also deepen an understanding of disease mechanisms and facilitate the development of novel treatment approaches. However, medical data are complex, often incomplete, and contains missing modalities, making its effective handling crucial for multimodal models training. We introduce **impuTMAE**, a novel transformer-based end-to-end approach with an efficient multimodal pre-training strategy. It learns inter- and intra-modal interactions while simultaneously imputing missing modalities by reconstructing masked patches. Our model was pre-trained on heterogeneous and incomplete data and subsequently fine-tuned for cancer survival prediction using The Cancer Genome Atlas (TCGA) dataset, integrating three data modalities: genetic data (DNA methylation and RNA sequencing), and imaging data (whole-slide histopathology images, WSI). By integrating missing data imputation directly into its pre-training phase, impuTMAE enables robust and resource-efficient learning, which establishes a new state-of-the-art for multimodal cancer survival prediction. Our code is available at `https://anonymous.4open.science/r/IMPUTMAE/`

## 1 Introduction

Today effective treatments remain unavailable for a large proportion of diseases — for instance, cancer, one of the most lethal and complex conditions World Health Organization (2024), which demands a comprehensive approach to both therapy and prognosis. One reason for this lies in their heterogeneous nature and individualized course of disease progression,as well as the large number of subtypes within a single disease. For instance, more than 50 histomorphological subtypes of lung cancer have been identified. Siddique et al. (2024). With the recognition of these features of disease manifestation and the pronounced variability among patients, new paradigms in medicine have emerged, such as precision medicine Ashley (2016) and personalized medicine Goetz & Schork (2018). Precision and personalized medicine approaches require the processing and analysis of large volumes of heterogeneous data for each patient, including genomic, clinical, molecular, or cellular characteristics, as well as other data types. Patient stratification into specific subgroups, the identification of targeted biomarkers, and the discovery of therapeutic agents represent non-trivial tasks that necessitate the integration and analysis of vast, heterogeneous datasets.

The most promising methodology for building models in precision and personalized medicine lies in deep learning approaches Grapov et al. (2018), which are capable of processing massive datasets and constructing effective models for a variety of predictive tasks. Among deep learning methods, a particularly important class is represented by multimodal deep learning models Yang et al. (2025a), which are able to effectively integrate heterogeneous data types and combine their representations to address subsequent predictive tasks in precision and personalized medicine. At the same time, training multimodal deep learning models in precision medicine entails a number of challenges. Despite the growing availability of biomedical data and the emergence of large-scale biobanks and datasets such as the UK Biobank , TCGA Tomczak et al. (2015), MIMIC Johnson et al. (2023) and others, the number of samples in such collections typically reaches tens or hundreds of thousands. In contrast, current state-of-the-art deep learning models — such as CLIPRadford et al.

(2021),FlamingoAlayrac et al. (2022), LLaVA Liu et al. (2024), or — are trained on millions of samples. So, a major challenge in medical data is the limited availability of large datasets comparable to those in language models and image processing, where deep learning excels. Expanding biobank resources is an extremely costly process, often requiring billions of dollars (e.g., [reference to biobank]), and many of the largest biobanks are either commercial (with access costs reaching substantial amounts) or restricted. Therefore, a critical challenge in biomedical deep learning is developing models robust to small datasets and missing modalities. Self-supervised learning presents a promising solution, as demonstrated by its effective application in studies such as LLaVA-MEDLi et al. (2023),MedCLIPWang et al. (2022), BiomedCLIPZhang et al. (2023)

The effective use of such data remains challenging, particularly in oncology, given the lethality and complexity of cancer. Within this field, accurate cancer survival prediction has emerged as a central task, as it is critical for patient stratification and treatment planning.

In our work, we focus on this task and *propose a novel multimodal pretraining strategy* based on the ViTMAE framework He et al. (2021), *which can effectively works with missing modalities.* Our multi-modal pretraining strategy integrates omics data with medical imaging (MRI and histology). The model is trained to reconstruct masked patches across all modalities simultaneously, thereby explicitly promoting the learning of cross-modal interactions during pretraining.

A key advantage of our approach over contrastive learning is that it does not require all modalities to be present for each sample. We treat any missing modality as a fully masked input and train the decoder to reconstruct it. This capability enables the robust integration of incomplete datasets, which is critical for leveraging real-world clinical data where modality scarcity is common. This flexibility and computational efficiency make our method a powerful solution for multimodal medical data analysis, addressing the limitations of previous techniques.

Furthermore, at the data integration stage, we address *the question of how to perform this process more effectively, taking into account data imbalance, noise, the presence of a dominant modality, and overall data limitations.* In addition, *we examine each modality individually, assess its contribution to survival prediction, and explain how this knowledge can be leveraged for more effective data integration.* Thus, our contributions are:

1. We introduce a unified, end-to-end multimodel model that integrates omics and medical imaging data. The model can be fine-tuned for diverse clinical tasks and is uniquely capable of imputing missing modalities.

2. We apply a novel ViT-MAE-based pretraining strategy tailored for multimodal data with a high proportion of missing modalities.

3. We quantified the relative contributions of each modality and tuned the fusion block to be robust to challenges of dimensionality, heterogeneity, and noise inherent in multimodal data.

4. Our model achieves state-of-the-art performance on multiple cancer survival analysis tasks, as a result of its universal integration strategy and robustness to missing data.

## 2 RELATED WORKS

### 2.1 SELF-SUPERVISED METHODS IN BIOMEDICAL ANALYSIS

Over the past few years, the adoption of self-supervised learning (SSL) has expanded rapidly in biomedical applications, especially for histopathology and omics. Liu et al. (2025) used a self-supervised Barlow Twins encoder to derive morphological features from colon cancer WSIs, clustering them into histomorphological phenotype clusters (HPCs) linked to patient survival and molecular–immune profiles. In a related effort, Divide-and-Rule Abbet et al. (2020) learns self-supervised representations of tissue patches, applies spherical k-means clustering, and aggregates cluster distributions and interactions into patient-level features for survival modeling, achieving significant prognostic stratification. Yang et al. (2025b) introduced BEPH, a BEiT-based foundation model pretrained with masked image modeling on 11.7 million unlabeled histology patches, and successfully transferred it to patch-level, WSI-level, and survival prediction tasks across cancers. However, most existing works are restricted to a single modality (typically imaging) or treat SSL only as a

pretraining step. In multimodal survival prediction, where imaging is combined with clinical or molecular data, self-supervision is rarely applied end-to-end. Instead, contrastive or reconstruction-based SSL are typically used separately per modality, after which the resulting feature vectors are fused for downstream survival modeling.

## 2.2 MULTIMODAL DL APPROACHES IN CANCER SURVIVAL PROGNOSIS

Recent multimodal deep learning frameworks for cancer survival prognosis show progress but also persistent gaps. Cheerla et al. Cheerla & Gevaert (2019) integrated four modalities using modality-specific encoders and multimodal dropout for missing data, whereas MultiSurv Vale-Silva & Rohr (2020) scaled to six modalities across 33 cancer types but handled missing inputs only through zero-vector imputation and simple substitutions. More advanced fusion was proposed by Zhou et al. Zhou & Chen (2023) with cross-modal translation and attention between histology and genomics, and by Gomaa et al. Gomaa et al. (2024) with a transformer combining self-supervised MRI pretraining and cross-attention for imaging, molecular, and clinical data. At MICCAI 2024, MoME Xiong et al. (2024) introduced biased progressive encoding with modality-specific experts, while PG-MLIF Pan et al. (2024) used gating attention and low-rank fusion for pathology–genomics integration. DRIM Robinet et al. (2024) disentangled shared and modality-specific encoders to enable robust survival prediction with incomplete modalities, handling missing data through masked attention fusion, whereas OmiCLIP Chen et al. (2025) was pretrained contrastively on a large-scale collection of histology–transcriptomics pairs, thus relying on paired data during foundation training. Collectively, these methods underscore the trade-off between robustness and scalability: some require all modalities to be present, others use simplistic imputation for missing data, and many lack broad pretraining or support only limited modalities. This motivates our work to explore whether multimodal pretraining enhances inference and cross-modal learning, improves patch-level reconstruction over unimodal baselines, and enables hybrid convolutional architectures for genomic data.

## 3 METHODS

**ImpuTMAE** is a multimodal end-to-end model based on a transformer encoder–decoder architecture (Fig. 1). The model is pretrained with a masked multimodal learning strategy, designed to reconstruct missing information across heterogeneous data sources. **In the pretraining stage**, 50% of the input patches from each of the modalities (RNA, DNAm, WSI) are randomly masked, and the encoder–decoder is trained to jointly reconstruct it. Moreover, under this training scheme we treat the missing modalities as fully masked, which enables the utilization of all available data during the pretraining stage. This objective forces the model to exploit complementary information across modalities, rather than relying solely on unimodal cues. **In the fine-tuning stage**, the pretrained network is leveraged for two purposes: (i) the decoder is employed for modality imputation, enabling the reconstruction of missing or corrupted modalities at inference time, and (ii) the encoder acts as a pretrained backbone for the downstream task of cancer survival prediction.

## 3.1 MODALITY-SPECIFIC ENCODERS

Such medical data is inherently heterogeneous, with each modality (RNA-seq, DNA methylation, WSI) exhibiting a distinct structure. To effectively process this variability, each data modality is handled by a dedicated encoder network, which transforms the raw input into a latent feature embedding of fixed size (denoted as $d$). We design these encoders with architectures specifically tailored to their respective data types, incorporating modality-specific design choices such as the number of layers, layer types (e.g., fully connected or convolutional), output feature dimensions, and key hyperparameters (e.g., activation functions, normalization, and dropout strategies). This modular design enables each encoder to extract high-level features from its respective input modality before these representations are fused within a multimodal decoder

**RNA-Seq Encoder (Gene Expression)** Firstly, The RNA-seq encoder embeds the preprocessed RNA sequence into a series of non-overlapping patches of size 512 by 1D convolution with added sinusoidal positional embeddings. After that it processes the resulting tokens via a series of 6 transformer blocks. Learnable CLS token are added to the patch embeddings. Subsequently, six transformer layers (more details in Table 1), similar to those in ViT-MAE, are applied to generate the

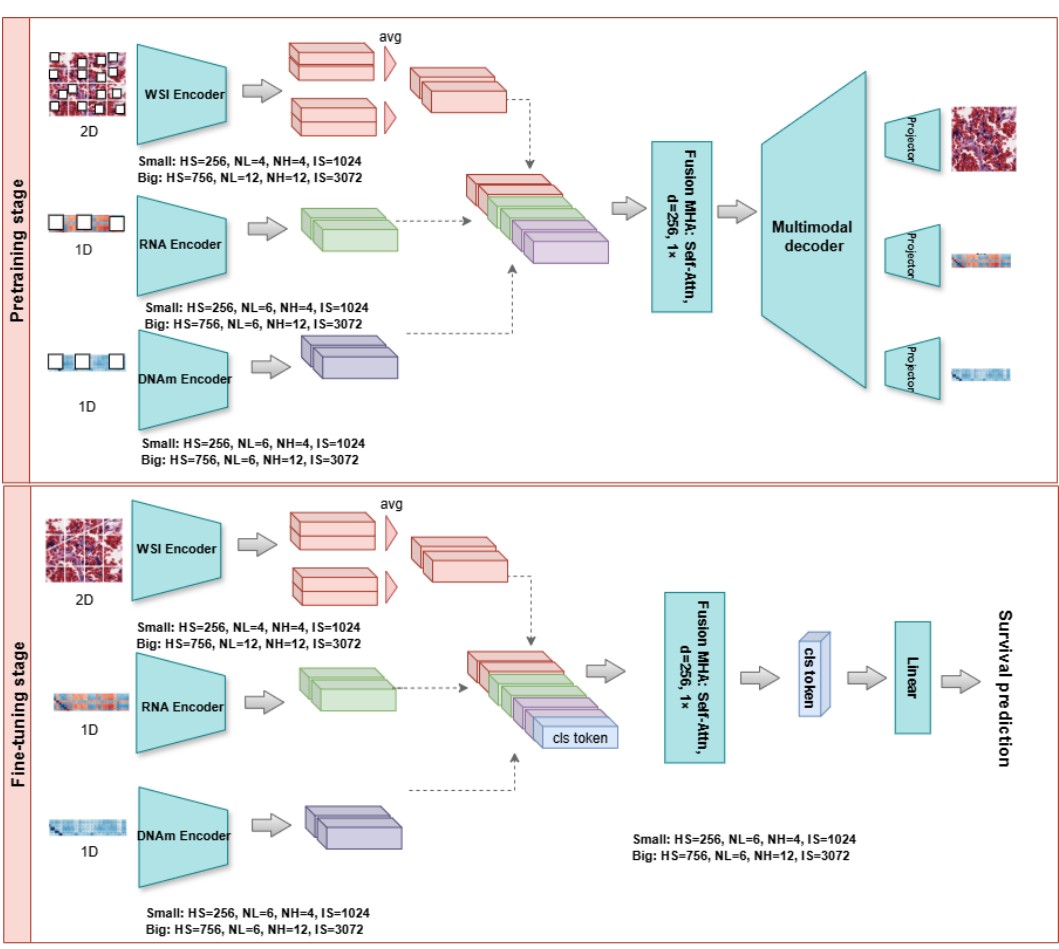

Figure 1: An overview of ImputMAE architecture, which includes modality-specific encoders for DNAm, RNA, WSI, and MRI, as well as a multimodal decoder that processes the concatenated representation to simultaneously reconstruct masked-out multimodal patches. For glioma survival analysis, we integrate a fusion multi-head attention block into the pretrained encoder model and utilize the decoder for imputing missing modalities.

hidden states $z_{\text{RNA}}$. Two configurations of the model were examined: a smaller variant with a hidden dimension of 256 and 4 attention heads, and a larger variant with a hidden dimension of 756 and 12 attention heads.

Table 1: Model hyperparameters for different modality-specific encoders.

| Encoder | hidden_size | num_layers | patch_size | num_heads | intermediate_size |
|---|---|---|---|---|---|
| DNA encoder | 256, 756 | 6 | 512 | 4, 12 | 1024,3072 |
| RNA encoder | 256, 756 | 6 | 256 | 4,12 | 1024,3072 |
| WSI encoder | 256, 756 | 4,12 | 16×16 | 4,12 | 1024, 3072 |
| Mult decoder | 256, 756 | 3,8 | to org | 4,16 | 512, 2048 |

**DNA Methylation Encoder** follows a similar design to the RNA-Seq Encoder, consisting of a single convolutional layer that partitions the DNA methylation array into non-overlapping patches with 512 size. This is followed by six transformer layers, akin to those in ViT-MAE, to produce the hidden representations $z_{\text{DNA}}$ .We considered two configurations with hidden representations of size 256 (small) and 756 (big).

**WSI Encoder (Histopathology Image)** splits a 256×256 patches into 16×16 patches, using a convolutional layer with a 16×16 kernel and stride 16, resulting in the hidden representations $z_{\text{WSI}}$. The patch embeddings are then augmented with positional encodings and a CLS token before being processed by a transformer.We evaluated architectures with hidden dimensions of 256 and 756. The 756-dimensional model comprised 12 transformer blocks and was initialized with pretrained `facebook/vit-mae-base` weights, while the lighter 256-dimensional variant with 4 transformer blocks was trained from scratch.

### 3.2 MULTIMODAL DECODER

The outputs from the individual encoders are concatenated into fused latent representation $[z_{\text{RNA}}, z_{\text{DNA}}, z_{\text{WSI}}]$ and fed to a multimodal decoder that reconstructs the original inputs for each modality. The multimodal decoder learns hidden representations through transformer layers (see Table 1), similar to ViT-MAE, and then projects these representations back to the original patch size of each modality using dedicated projection heads. The projection head for WSI consists of a 2D transposed convolution layer, and for RNA and DNA—linear layers. A reconstruction loss (mean squared error) is computed between the original data and the reconstructed data on the masked patches for each modality. The final multimodal MSE loss across all $M$ modalities is computed as:

$$\text{MSE}_{\text{multimodal}} = \sum_{m=1}^{M} \text{MSE}^{(m)}, \tag{1}$$

$$\text{MSE}^{(m)} = \frac{1}{N_m} \sum_{i=1}^{N_m} \left( y_i^{(m)} - \hat{y}_i^{(m)} \right)^2, \quad m = 1, 2, \ldots, M. \tag{2}$$

### 3.3 FINE-TUNING FOR SURVIVAL ANALYSIS

**Loading and Freezing Pretrained Modality-Specific Encoders:** We employ the pretrained encoders—originally optimized jointly with the multimodal decoder—as feature extractors for subsequent fine-tuning. During this stage, only the final layer of each encoder is unfrozen, while the fusion block and all preceding layers remain fixed.

**Handling Missing Modalities:** The pretrained multimodal decoder is employed to impute missing modalities, ensuring a consistent multimodal representation.

**Fusion multi-head attention block:** The embeddings generated by each modality-specific encoder are concatenated into a unified multimodal representation. For WSI data, all patch embeddings are aggregated by averaging their CLS tokens before concatenation to construct the final patient-level representation. Finally, Fusion multi-head attention block is applied to effectively integrate heterogeneous features. This block consists of a single self-attention layer with a latent size of either 256 or 756, depending on whether the larger or the smaller encoders are used.

**Survival Risk Projection:** The fused multimodal representation is processed through a final projection layer (a linear layer), which maps the multimodal embedding to survival risk scores across $T$ time intervals. Following prior work Robinet et al. (2024), we set $T = 20$.

In survival analysis, we use a negative log-likelihood (NLL) loss Gorgi Zadeh & Schmid (2021) based on hazard logits, defined as:

$$\mathcal{L} = -\frac{1}{n} \sum_{i=1}^{n} \sum_{t=1}^{\kappa(t_i)} \left( y_{it} \log h(\tau_t | x_i) + (1 - y_{it}) \log(1 - h(\tau_t | x_i)) \right), \tag{3}$$

where $h(\tau_t | x_i)$ is the discrete-time hazard function, modeling the probability of an event at $\tau_t$ given prior survival. The indicator $y_{it}$ is 1 if the event occurs at $t$, otherwise 0. The loss is averaged over $n$ patients.

## 4 EXPERIMENTS

We evaluated our approach on multimodal cancer datasets from The Cancer Genome Atlas (TCGA) Chang et al. (2013), focusing on the three largest cohorts with multimodal data: Glioma (GBM + LGG, $n = 1104$; GBM=592, LGG=512), Breast invasive carcinoma (BRCA, $n = 985$), Uterine corpus endometrial carcinoma (UCEC, $n = 491$). GBM and LGG were analyzed jointly as the TCGA Glioma cohort, reflecting their continuum of disease progression and to improve statistical power while capturing both shared and grade-specific signals.

For each cohort, we integrated five complementary data modalities:

- **Gene expression (RNA-seq):** 16,304 protein-coding genes (variance $> 0.1$) were retained, FPKM-UQ normalized, log-transformed, and patient-wise normalized to reduce gene-specific noise while preserving inter-sample variability Bullard et al. (2010). Since the data distribution exhibited a long-tail pattern, we first applied upper-quartile normalization, scaling each sample (row) by dividing its values by the 75th percentile. Subsequently, a Torch QuantileTransformer was employed to discretize the values into 10,000 quantiles, followed by min–max normalization to ensure consistent scaling across features.

- **DNA methylation (DNAm):** 25,978 Beta values processed following Robinet et al. (2024). The same procedure with upper-quartile normalization, quantile discretization and min–max normalization as in RNA-seq.

- **Histopathology (WSI):** For each patient, one artifact-free H&E-stained slide was down-sampled and tissue-masked using OTSU Otsu (1975). From 1000 randomly sampled $256 \times 256$ patches, we selected those with the highest HSV intensity.

Table 2 summarizes per-cohort coverage of DNA methylation, clinical, and histopathology data among patients with available RNA-seq.

| | Modality availability (%) | | | |
|--------|------|----------|-------|-----------|
| Cohort | DNAm | Clinical | WSI | RNA count |
| BRCA | 99.69 | 91.24 | 96.44 | 982 |
| UCEC | 99.59 | 95.30 | 92.43 | 489 |
| GBM | 80.71 | 97.14 | 68.21 | 280 |
| LGG | 99.22 | 93.55 | 94.92 | 512 |

Table 2: Modality availability (%) across TCGA cohorts among patients with RNA-seq data.

This multimodal integration reflects real-world heterogeneity in data availability while maximizing the complementary strengths of molecular, imaging, and clinical modalities.

**Implementation Details** At the *multimodal masked pretraining* stage, ImputMAE was trained on different modality subsets using the AdamW optimizer (lr $= 1 \times 10^{-4}$, wd $= 1 \times 10^{-2}$) for

800 epochs with a linear warmup over the first 50 epochs and a batch size of 256. . We use several pretraining setups with 256 hidden size and 756 hidden size (initialized with pretrained weights from `facebook/vit-mae-base`). During the *fine-tuning* stage for survival prognosis prediction, the models were trained for 20 epochs with a dropout rate of 0.1 applied to each modality subset, a batch size of 24, a weight decay of $1 \times 10^{-2}$, and a linear warmup during the first 5 epochs and early stopping on best c-index. For all experiments, the learning rate was fixed at $1 \times 10^{-4}$. We employed the larger model with a hidden size of 756 for the GBM and LGG datasets, whereas the smaller model with a 256-dimensional hidden representation was used for the smaller datasets (BRCA, BLCA, and UCEC).

**Evaluation** For evaluation, we split the dataset into 80% training and 20% testing sets. The best hyperparameters were selected via five-fold cross-validation, and models were assessed on the test set. We use the *Concordance Index* (C-index) Antolini et al. (2005), a widely used metric for ranking survival times. However, when evaluating methods on a several number of modalities, the number of samples containing all modalities decreases dramatically. To address this, we evaluate our approach on the entire dataset where at least the RNA modality is available thus conduct more robust comparison. We impute missing modalities in prior approaches using the proposed method in the article. In our approach, we impute missing modalities with pretrained multimodal decoder.

## 5 RESULTS

Table 3: Comparison of C-index for survival prediction across different TCGA cohorts.

| Model | Modality | BRCA | UCEC | GBM+LGG |
|---|---|---|---|---|
| MoME | all | 0.604+-0.066 | 0.640+-0.058 | 0.846+-0.020 |
| CMTA | all | 0.478+-0.07 | 0.539+-0.045 | 0.829+-0.011 |
| MultiSurv | all | 0.498+-0.057 | 0.579+-0.111 | 0.781+-0.107 |
| impuTMAE | RNA | **0.610+-0.044** | 0.620+-0.030 | **0.863+-0.009** |
| impuTMAE | DNAm | 0.512+-0.036 | 0.496+-0.009 | 0.839+-0.016 |
| impuTMAE | WSI | 0.499+-0.043 | 0.5+-0.04 | 0.636+-0.02 |
| impuTMAE | all | 0.592+-0.04 | **0.75+-0.032** | **0.870+-0.003** |

We compared our proposed method impuTMAE with previous state-of-the-art approaches, and the results are presented in Table 3. As we see, our method outperforms the vast majority of prior works in multimodal or unimodal settings, demonstrating that a transformer-based approach with multimodal pretraining and missing modality imputation is one of the most effective solutions for survival prediction. Unlike previous studies, which reported performance based on cross-validation with early stopping, we conducted a rigorous evaluation on five held-out folds. As a result, the reported metrics may be lower than those presented in the aforementioned works.

Additionally, our findings corroborate previous research Vale-Silva & Rohr (2020), Xiong et al. (2024), as shown in Table 3, confirming that RNA is the most critical modality for survival prediction in cancer. In addition, we evaluated the reconstruction performance of multimodal decoders in combination with unimodal encoders for each modality 4. Our findings show that, given the same number of training epochs, reconstruction quality in the multimodal setting improved substantially for RNA expression and showed consistent gains for DNA methylation. For histopathology, further comparisons are required, as both the encoder and decoder were initialized with pretrained weights from facebook/vit-mae-base; consequently, in this case the unimodal setup demonstrated higher reconstruction performance (see Table). Results show MSE across all datasets on test split.

Table 4: Reconstruction performance (MSE) on on the held-out test set across modalities under unimodal and multimodal pretraining.

| Pretraining | WSI | RNA | DNAm |
|---|---|---|---|
| Multimodal pretraining | 0.028 | **0.0039** | **0.0085** |
| Unimodal pretraining | 0.0165 | 0.0092 | 0.0087 |

## 6 CONCLUSION

We present an end-to-end unified multimodal framework that employs a multimodal masked pre-training strategy, enabling both fine-tuning for diverse downstream tasks and the imputation of missing modalities. Our experiments demonstrate that this pretraining paradigm substantially enhances multimodal representations, yielding superior performance in both masked reconstruction and downstream prediction tasks. These advances enable the proposed method to achieve state-of-the-art results in cancer survival prediction. Furthermore, the multimodal masked pretraining approach maintains robust performance in the presence of incomplete input data, an essential property for clinical and biomedical applications where missing modalities are frequently encountered.

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
