# OpenReview forum: "impuTMAE: Multi-modal Transformer with Masked Pre-training for Missing Modalities Imputation in Cancer Survival Prediction"
_ICLR.cc/2026/Conference — Submitted to ICLR 2026_

### Official Review · Reviewer_SUvc · 2025-10-15

**Soundness:** 3
**Presentation:** 2
**Contribution:** 2
**Rating:** 4
**Confidence:** 3

**Summary:**

This paper introduces ImputMAE, a unified, end-to-end transformer-based framework for multimodal cancer survival prediction that is specifically designed to handle incomplete datasets. The core of the approach is a multimodal masked pre-training strategy inspired by Masked Autoencoders (MAE). The model is pre-trained to reconstruct randomly masked patches from multiple modalities simultaneously (e.g., RNA-seq, DNA methylation, and WSI). Critically, an entire missing modality for a given sample is treated as "fully masked," compelling the model to learn deep inter-modal dependencies to impute it from the available data. After pre-training, the powerful encoder serves as a feature extractor for the downstream survival prediction task, while the decoder can be used to perform explicit modality imputation at inference time.

**Strengths:**

1. Unified Framework for Imputation and Representation Learning: The most significant strength is the elegant integration of missing modality imputation directly into the self-supervised pre-training objective.

2. Addressing a Critical Real-World Challenge: The model is explicitly designed to work with incomplete datasets, which is a pervasive problem in clinical and biomedical research.

3. Efficient Pre-training Strategy: By building on the MAE framework, the model inherits its computational efficiency. The encoder only needs to process the visible (unmasked) patches, which significantly reduces the computational load during the expensive pre-training phase compared to contrastive learning methods that must process all inputs.

**Weaknesses:**

1. There are relatively few baselines for comparison, and most are not from 25 years ago.

2. There are only three TCGA datasets for testing.

3. Kaplan–Meier curve analysis can provide a better credibility and understandability of the model.

4. How much does the performance, particularly for the WSI modality, depend on the external pre-trained weights versus the ImputMAE framework itself?

**Questions:**

1. There are relatively few baselines for comparison, and most are not from 25 years ago.

2. There are only three TCGA datasets for testing.

3. Kaplan–Meier curve analysis can provide a better credibility and understandability of the model.

4. How much does the performance, particularly for the WSI modality, depend on the external pre-trained weights versus the ImputMAE framework itself?

---

### Official Review · Reviewer_yNWj · 2025-10-27

**Soundness:** 2
**Presentation:** 2
**Contribution:** 2
**Rating:** 2
**Confidence:** 5

**Summary:**

This paper introduces impuTMAE, a multimodal transformer framework designed to address the pervasive challenge of missing data modalities in cancer survival prediction. Built upon the Masked Autoencoder (MAE) scheme, the model integrates three data types: genomic (RNA-seq, DNA methylation) and histopathological (Whole-Slide Images, WSI). It adopts a unified pre-training strategy where patches from all modalities are randomly masked, and the model is tasked with reconstructing them. After pre-training, the model is fine-tuned for survival prediction. The encoder serves as a feature extractor, with a fusion attention block integrating the modalities, while the decoder is used for imputing any missing data at inference time. The method is rigorously evaluated on three TCGA cancer cohorts (Glioma, BRCA, UCEC). Results demonstrate that impuTMAE achieves improved performance in survival prediction (measured by C-index) and shows superior reconstruction quality for genomic data in the multimodal setting compared to unimodal baselines. However, the paper seems to have been prepared in a rush and is obviously not ready for publication.

**Strengths:**

+ The paper directly tackles a critical, real-world problem in medical AI: incomplete multimodal datasets.
+ The model is tested on multiple cancer types with varying data availability.

**Weaknesses:**

- The technical innovation in the presented work is limited. MAE-based pre-training followed by fine-tuning for survival analysis is not a new training paradigm.
- While the model demonstrably improves reconstruction and prediction, the paper offers limited analysis into how the cross-modal interactions occur.
- Results of baselines (e.g., direct fine-tuning without pre-training) should be included. Results of SOTA methods for the missing modality should be included in the comparison study.
- It is not clear how the pre-trained model can benefit the downstream tasks in comparison to other pre-trained models or domain models.

minor:
- (e.g., [reference to biobank]) on line 057
- The reference style is not correct, e.g., as LLaVA-MEDLi et al. (2023), MedCLIPWang et al. (2022), BiomedCLIPZhang et al. (2023)
- What's MRI on line 203?
- What are "five complementary data modalities"? on line 293

**Questions:**

see weakness

---

### Official Review · Reviewer_JQna · 2025-10-30

**Soundness:** 1
**Presentation:** 1
**Contribution:** 2
**Rating:** 2
**Confidence:** 5

**Summary:**

This paper presents impuTMAE, a novel end-to-end, transformer-based framework for cancer survival prediction using multimodal data. Its core contribution is a multimodal masked pre-training strategy based on the MAE framework, which is designed to learn both intra- and inter-modal interactions. A key feature is its capability to handle missing modalities by treating them as fully masked inputs; thus, it learns to impute them during pre-training. The model is pre-trained to reconstruct masked patches across all modalities and is subsequently fine-tuned for survival prediction.

**Strengths:**

1. The core idea of extending the masked autoencoder (MAE) paradigm to a heterogeneous multimodal setting (i.e., genomics and imaging) is a non-trivial and valuable contribution.


2. This work addresses an important problem in medical AI—clinical datasets are almost always incomplete.

**Weaknesses:**

1. The paper's state-of-the-art claim is unreliable due to a critical flaw in the experimental setup. As explicitly stated in Section 4, the authors apply their own proposed imputation method to all prior approaches. This invalidates the comparison in Table 3. The experiment does not compare impuTMAE against the original MOME or MultiSurv, but rather against those models modified and augmented with impuTMAE's imputer. This makes the reported SOTA results unreliable.


2. In Table 3, the full multimodal impuTMAE (all) model achieves a C-index of 0.592 on BRCA, which is not only lower than the MOME baseline (0.604) but also substantially worse than the authors' own unimodal impuTMAE (RNA) model (0.610). This strongly suggests that multimodal fusion is detrimental for this cohort, a fact the authors do not address. Moreover, the comparison is missing several recent and highly relevant SOTA models in multimodal survival analysis from 2024-2025, such as:

    [1] Multimodal Prototyping method for cancer survival prediction.

    [2] From Single-Cancer to Pan-Cancer Prognosis: A Multimodal Deep Learning Framework for Survival Analysis with Robust Generalization Capability.


3. The methodology for handling WSI (Whole Slide Imaging) is overly simplistic and ignores established best practices in the computational pathology field. HSV intensity metrics are simple color-space measures and have no established correlation with tumor biological or prognostic features.


4. Results in Table 4 contradict its central hypothesis. Multimodal pre-training results in a significantly higher WSI reconstruction MSE than unimodal pre-training. The authors attribute this to the unimodal model using pre-trained weights but do not explain why the joint objective does not at least match the unimodal performance. This result suggests that multimodal pre-training may actually be harmful to WSI feature learning.


5. The paper suffers from multiple formatting issues, including low-resolution figures, a missing citation on Line 058, and confusing ± symbols in Table 3.

**Questions:**

Please refer to the **Weaknesses**.

---

### Official Review · Reviewer_3U1X · 2025-10-31

**Soundness:** 2
**Presentation:** 2
**Contribution:** 2
**Rating:** 2
**Confidence:** 4

**Summary:**

The paper presents a model called inpuTMAE to address the challenge of missing modality during cancer survival prediction by pretraining the reconstructing models on different modalities. The experimental results somewhat demonstrate the effectiveness of this method, but there are no ablation studies, and the paper framing has serious issues.

**Strengths:**

1.	The idea of using MAE to address missing modality challenge is suitable and straightforward

2.	The experimental results are proving the effectiveness of this method.

**Weaknesses:**

1.	The evaluation is not convincing due to the fact that the comparing baselines in this paper are severely limited. There are plenty of recent survival prediction methods, and the authors only choose MoME, CMTA, MultiSurv. In addition, the authors should consider MOTCAT, MCAT, PIBD, SurvPath, MMP, etc. Specifically, in CVPR2025, DisPro addresses the missing modality challenge in survival prediction, which the authors should compare their methods with.

2.	There are no ablation studies on the components of inpuTMAE, like the number of layers, patch size, hidden size, positional embedding, different architectures, etc.

3.	There are also formatting issues. \citep{xxx} should be used in most of the cases in this paper, but the authors used \citet{xxx}. The commas should be separated with the latter word.  Besides, the paper is even only 7 pages.

4.	Terminologies used in this paper is strange. MAE paper is usually called “MAE”, not “ViTMAE”, and why MAE works well with missing modalities (at least the original paper does not say so, and if the authors insist on this, please cite relevant papers)?

5.	The authors claim in line 068 that they pretrain a model using omics with histology and MRI, however, in the figure and section 3, it seems the authors only pretrain the model with omics and histology, no MRI involved.

6.	The organization of this paper is not good. The first paragraph in the introduction is lengthy and does not contain much information related to the main content, the proposed method. Also, the whole paper reads like a technical report, not a scientific paper that brings ideas. There are plenty of technical details, but not insights.

7.	This pretraining can be computational heavy. The authors are suggested to report the time and GPU devices used for it.

**Questions:**

1.	Even if the missing modalities are treated as all masked modalities, how can the model determine whether it generates the right or wrong modality, i.e., how to calculate the loss? I understand the authors have written down the loss calculation, but the prerequisite of that equation is the present of all modalities.

2.	Why choose 50% as the mask ratio? Are there any empirical evidence or relevant papers supporting this?

---

### Meta-Review · Area_Chair_QPex · 2026-01-02

**Summary:**

The manuscript proposes impuTMAE, a transformer-based model to handle modality missingness. The method is tested on TCGA data with three modalities: DNA methylation, RNA sequencing, and whole-slide imaging. While reviewers found the problem studied interesting and relevant to the community and the MAE-based framework is reasonable, they raised quite a few major concerns regarding technical contribution and experiments:

1. MAE-based pre-training followed by fine-tuning for survival analysis is not a new training paradigm, which limits the technical novelty and contribution.
2. Many SOTA methods are missing (see reviewers' comments for more details).
3. Writing and organization should be improved, to be more rigorous, and smoother. More insights are needed instead of minor technical details.
4. Computational cost (time and space) should be reported for fair comparison.

**Reviewer Concerns:**

The authors didn't respond to any reviewer, so all concerns are still valid.

**Reviewer Scores:**

The authors didn't respond to any reviewer, so I don't think the scores would have been changed.

---

### Decision · Program_Chairs · 2026-01-26

Reject